# How Do Health, Biological, Behavioral, and Cognitive Variables Interact over Time in Children of Both Sexes? A Complex Systems Approach

**DOI:** 10.3390/ijerph20032728

**Published:** 2023-02-03

**Authors:** Elenice de Sousa Pereira, Mabliny Thuany, Paulo Felipe Ribeiro Bandeira, Thayse Natacha Q. F. Gomes, Fernanda Karina dos Santos

**Affiliations:** 1Department of Physical Education, Federal University of Viçosa, Viçosa 36570-900, MG, Brazil; 2Faculty of Sports, University of Porto, 4200-450 Porto, Portugal; 3Department of Physical Education, Regional University of Cariri—URCA, Crato 63105-000, CE, Brazil; 4Federal University of Vale do São Francisco—UNIVASF, Petrolina 48902-300, PE, Brazil; 5Department of Physical Education, Federal University of Sergipe, São Cristóvão 49100-000, SE, Brazil; 6Department of Physical Education and Sport Sciences, University of Limerick, V94 T9PX Limerick, Ireland; 7Physical Activity for Health Cluster, Health Research Institute, University of Limerick, V94 T9PX Limerick, Ireland

**Keywords:** growth and development, fundamental motor skills, physical activity, health, cognition, complex systems, childhood

## Abstract

The present study examined gender differences in health, physical activity, physical fitness, real and perceived motor competence, and executive function indicators in three time points, and analyzed the dynamic and non-linear association between health, biological, behavioral, and cognitive variables in children followed over time. A total of 67 children (aged between six and 10 years) were followed during two years and split into two cohorts (six to eight years old: C1; eight to 10 years old: C2). Data regarding health, physical activity, real and perceived motor competence, physical fitness, and executive function indicators were obtained according to their respective protocols. Comparison tests and network analysis were estimated. Significant gender differences were found in both cohorts. The emerged networks indicated different topologies in both cohorts. No clusters were observed between the variables in C1, and there was a greater number of interactions at eight years of age. Sparse networks were observed in children aged eight and 10 years in C2, and greater connectivity was observed at nine years of age between health, physical fitness, motor competence, and physical activity indicators. This study showed that there are non-linear dynamic relationships between health, biological, behavioral, and cognitive variables over time during child development.

## 1. Introduction

Children not only undergo changes in physical/biological aspects during the growth and development processes, but also in cognitive, social, and behavioral aspects, which can echo throughout life [1,2]. Physical activity stands out among the variables associated with these changes, and involves a wide range of movements considered essential for independence and interaction with the environment, such as functionality, performance, leisure, and well-being [1,3]. The interaction between movement activities and the environment favors performing motor skills, as well as the underlying mechanisms (motor control and coordination) referred to in the literature as motor competence [4,5,6,7,8], which has been highlighted as important for an active and healthy lifestyle with relevance to child growth and development [9,10,11].

In addition to physical activity and motor competence, body composition [12], perceived motor competence [13], physical fitness [14], and executive functions [15,16] have been identified as important factors for children’s health. In this context, Bremer and Cariney [17] synthesized the literature that examined the impact of movement skills in five health areas (physical activity, physical fitness, body composition, self-perception, and executive functioning), and the results highlight that there is evidence that movement ability can have a positive influence on broad health domains, both over developmental time and through interventions. Similar results were reported in other studies [5,12,18]; however, given the complexity, the mechanisms by which all these variables interact and cooperate to generate healthy movement and behavior patterns in children still seem to be unknown, especially when it comes to different age groups.

There are many studies that have paid attention to understanding the factors (biological, behavioral, and cognitive) associated with human growth and development in both genders [12,16,19]. A general point in common among the works that investigated the relationships between physical activity, real and perceived motor competence, physical fitness, executive function, and health indicators in children is to point out the differences between the genders [20,21], in addition to considering that these relationships are established linearly [13], disregarding dynamic and non-linear synergistic interactions between variables. In this perspective, cross-sectional studies [22,23] have shown that there are non-linear relationships between movement behavior variables, fundamental motor skills, and screen time in preschoolers (as far as we know, studies that have explored the same relationships in children aged six to 10 are still unknown). On the other hand, longitudinal studies, which can describe in depth the changes resulting from the growth and development process and allow the observation of the non-linear interactions between health, biological, behavioral, and cognitive variables, are scarce especially in second childhood [24]. For example, Cazorla-González et al. [24] explored the impact of crawling before walking on a network of interactions between body composition, cardiovascular system, lung function, motor competence, and physical fitness in children seven years old and evaluated the longitudinal association between the studied variables. The authors observed that the group that crawled before walking presented more links among all the variables studied compared to the group that did not crawl. In this sense, new investigations (cross-sectional, longitudinal, and intervention) are needed to try to fill to this gap.

Since human behavior presents emergent properties as a result of the interaction between different variables [25,26], the use of dynamic systems and complex systems theory can help to better understand the phenomena of change and clarify how behavior (of a given and/or a set of variables) changes over time [27]. Furthermore, the system itself has an intrinsic tendency to create certain patterns [27], which can result in different network configurations and provide useful information about the specific role of components within an integrated network [28]. Thus, the relationship between health, physical activity, physical fitness, real and perceived motor competence, and executive function indicators can be better understood through network science [25], rather than just being examined by an analysis of associations separately, as in comparison analyses, which, notwithstanding their relevance, do not allow a deeper understanding about the interactions between multiple variables. Assuming that these variables are mutually related during childhood, and the interactions between them are complex, and that different patterns of networks can emerge, understanding how interactions occur over time can provide relevant answers to existing gaps on the subject [29,30]. In addition, understanding these emerging network patterns can provide insights for future interventions in the most sensitive variables or set of variables in the system. The main objective of this study was to analyze, through network analysis, the dynamic and non-linear association between health, biological, behavioral, and cognitive variables in children monitored over time, and as a secondary objective, to estimate the gender differences in health, physical activity, physical fitness, real and perceived motor competence, and executive function indicators in three moments.

## 2. Materials and Methods

### 2.1. Ethics Declaration

This study was conducted in accordance with the guidelines of the Declaration of Helsinki and all procedures were approved by the Ethics Committee in Research involving human beings of the Federal University of Viçosa, under process number 1.888.177.

### 2.2. Study Design and Participants

This longitudinal study is part of the research project “Relationship between Physical Activity, Motor Competence, Cognitive Skills and School Performance in Children and Adolescents from three to 12 years old”. The participants were monitored over time and evaluated at three different times one year apart (3 evaluations over 3 years). Thus, the first evaluation (baseline, T0) was performed between February/March and July/August 2017, followed by an evaluation in 2018 (T1), and another in 2019 (T2). Children of both genders aged between six and 10 years, regularly enrolled between the first and fifth grades of the elementary school in the only public elementary school in the city of Santo Antônio do Grama, Minas Gerais, Brazil, were sampled. In the last census, the municipality had a demographic density of 31.37 inhabitants/km^2^ and an average population of 4.085 inhabitants [31].

A total of 113 children were enrolled in the first and third years of an elementary school in the baseline. Of this total, only 89 children agreed to participate in the study. Of the 89 children invited to participate in the study, 22 were excluded (not having complete data in the three moments). A total of 67 healthy school-aged children (41 boys and 26 girls; six to 10 years old) were included in a longitudinal cohort study. The children (N = 36) from cohort 1 started the study at the age of six years and completed it at the age of eight years in the last year of assessment (T2). The children (N = 31) from cohort 2 started the study at the age of eight years and completed the research at the age of 10 years. Subjects were recruited from the same school.

The inclusion criteria were: (1) children between six and 10 years of age; enrolled at their first and third grades of elementary school; (2) being regularly enrolled in school; (3) voluntarily agreeing to participate by the Informed Consent Form signed by the legal guardians; and (4) not presenting with physical and/or cognitive disability. Exclusion criteria were: (1) not having information on all the variables analyzed for the three years of evaluation.

### 2.3. Instruments and Procedures

#### 2.3.1. Health Indicators

Waist-to-height ratio (WHtR) and body fat percentage (BFP) were included as health indicators. WHtR was obtained by dividing waist circumference (cm) by height (cm). Height was measured using a portable stadiometer (Sanny^®^, Brazil), and waist circumference was taken at the narrowest point between the lower costal margin and the iliac crest with a measuring tape (Sanny^®^, Brazil) with a precision of 0.1 cm [32]. The BFP was calculated from the triceps and subscapular skinfolds [33], and measured using an adipometer (Mitutoyo, BGF308, Cescorf^®^).

#### 2.3.2. Physical Activity

Physical activity was measured by applying a questionnaire and using pedometers. Baecke’s questionnaire on habitual physical activity [34] was applied to children through direct interviews by the evaluators. The questionnaire consists of 16 questions that are distributed in three different sections and aims to establish estimates of the habitual physical activity level of children. The answers are coded on a five-point Likert scale, except for questions 1 and 9. Scores are obtained based on the answers to the questions grouped in each of the sections equivalent to: occupational physical activity (practices physical activity at school/job); sporting physical activity (related to sports, and physical exercise); and leisure (occupation of free time), in addition to estimating the total score of habitual physical activity, with the latter being considered in the analysis.

The children used pedometers (Yamax, Digi-Walker, SW 200) for eight consecutive days, including two weekend days, attached to their waist in the right mid-axillary line. They were instructed to wear step monitors during their waking time and should only remove them for water activities such as showering and swimming. The children’s legal guardians received a diary for using the monitor, in which they should write down the time the device was put on, taken off, and the number of steps recorded on the device’s display daily. The children should have used the device with records of the number of steps on at least four days of the week, one of which should be on the weekend, to be considered “eligible” to compose the study sample [35]. The average value of the number of steps was considered in the analysis.

#### 2.3.3. Motor Competence

Motor competence was determined by the Gross Motor Development Test–2 (TGMD–2) [36], consisting of six locomotion skills (running, leaping, hopping, horizontal jump, galloping and sliding) and six object control skills (catching, throwing, kicking, hitting, dribbling, and rolling). Each skill was described and demonstrated once by the evaluator, followed by two tests performed by the child, according to the protocol for administering the TGMD–2 [36]. Video footage of each skill has been edited into single movie clips. In the analysis, each skill was performed twice, and each attempt was scored on the skill criteria as either successful (value 1) or unsuccessful (value 0). The scores from the two trials were added together to obtain a raw score for each skill. Scores for all skills were summed (total skill score range 0–96, locomotor 0–48, object control 0–48). Two evaluators were qualified and trained for the analysis of each of the skills. The inter-rater reliability in locomotion and object control skills was 0.75 and 0.98, respectively.

#### 2.3.4. Physical Fitness

Physical fitness was estimated by evaluating different components (handgrip strength, speed, agility, lower limb strength, upper limb strength, abdominal resistance, flexibility, and cardiorespiratory fitness) from different protocols (EUROFIT, FITNESSGRAM, AAHPERD, and PROESP-BR).

Handgrip strength: measured with a digital dynamometer (JAMAR^®^, model 5030, J1). The child should be in a standing position using their dominant hand, arm extended and slightly away from the body, and should press the device with maximum force at the evaluator’s command. The same procedure was performed with the non-dominant hand. Two attempts were recorded for each hand, with the highest value being recorded. The average of the best results of each hand (dominant and non-dominant) was used to compose the analysis [37].

Speed: estimated by the Shuttle run 10 × 5 test. Two parallel lines, five meters apart from each other, were drawn. The children had to run as fast as possible between one line and another, crossing it ten times. At the end of the last lap, the time (in seconds) of each child was recorded [37].

Agility: based on the square test, which consists of walking a delimited course of four × four meters as quickly as possible. The time (in seconds) spent on the test by the child was computed [37].

Abdominal muscular resistance: measured by the curl-up test [38]. The child started in the dorsal decubitus position with their hands positioned behind their neck, knees flexed (approximately 90 degrees), feet flat on the ground, and they should return to the sitting position, taking their elbows forward to touch their knees. The child should perform the highest number of repetitions possible in 30 s.

Upper limb strength: estimated from the push-up test [38]. The child started in the ventral decubitus position, keeping their elbows extended, hands resting on the floor and in line with their shoulders, and they should perform as many flexions as possible at approximately 90° degrees and elbow extensions for as long as they could perform the movement correctly.

Lower limb strength: evaluated from the horizontal jump test. The children were initially positioned behind the starting line with their feet parallel to the ground, and they performed a horizontal jump as far as possible, allowing the movement of arms and torso. The child should finish executing the movement with both feet on the ground and without losing balance. The distance was recorded between the starting line and the closest point of contact with the ground to this line. Two attempts were recorded, with the average value considered in the analysis [39].

Flexibility: evaluated by the sit and reach test [40]. The child started by sitting on the floor with their legs extended, without shoes, and should reach the maximum distance possible they could on a wooden bench (Wells bench) with both hands by flexing their trunk. Two trials were computed, and the mean value of both trials was used in the analysis.

Cardiorespiratory fitness: measured from the six-minute running and walking test [39]. The test consists of running and/or walking for six minutes on a defined course (18 × 9 m). The result was determined by the final distance (in meters), considering two decimal places. The raw value of the distance was considered in the analysis.

#### 2.3.5. Perceived Motor Competence

The Pictographic Assessment Scale of Perceived Motor Skills Competence for Children [41] was used to determine perceived motor competence. It consists of 12 pictographic tasks (six tasks related to locomotion skills and six related to object control skills), in which perception in each skill is evaluated from one to four points (four points represent high perception). The perception evaluation process for each skill uses a double and dichotomous process (meaning the child should first point to which of the images best represents them: the picture of a child who is competent in a skill or another child who is not so competent in a skill). Then, in line with the previous choice (competent or not), the children must again choose between two options: for the competent child (four points are assigned for ‘really good at’ or three points for ‘very good at’) and for the child who is not as competent (‘good at’ is awarded two points or ‘not so good’—one point). The total score of the scale can range from 12 to 48 points. Higher values denote greater perceived motor competence.

#### 2.3.6. Executive Functions

Central executive functions were assessed, including inhibitory control, working memory, and cognitive flexibility.

Inhibitory control: The Five Digits Test (FDT) was used to measure inhibitory control [42]. The FDT is a numerical neuropsychological task used to assess the Stroop effect, divided into four components (reading, counting, choosing, and changing). The first two components encompass measures of automatic attention and processing speed. The third component involves a selective attention test (inhibitory control), being estimated in time (seconds). The last component focuses on executive attention (or top-down attentional control). The raw score of the third component, which refers to inhibitory control, was used in the analysis. Higher scores indicate worse performance [42].

Working memory: the Digits test (subtest of the Wechsler Intelligence Scale for Children–Fourth Edition–WISC IV) [43] was used to assess working memory. The test consists of reproducing eight sequences of digits in two basic orders: (i) direct order (example of a sequence, 1-4-8); and (ii) reverse order (example of a sequence, 8-4-1). The participant has two attempts in both sortings, and there is a gradual increase in the number of digits for each sequence. Higher scores indicate better performance.

Cognitive flexibility: the Alternating Semantic Verbal Fluency test was employed to assess cognitive flexibility [44,45]. The verbal fluency task consists of producing the largest number of semantic words in the Animals and Fruits categories in 60 s. The number of words produced correctly, incorrectly, and corrected were computed. The test evaluates information processing speed and requires a focused search in memory (executive functioning). Higher scores indicate better test performance.

### 2.4. Data Collection Procedures

First, the signatures of the Informed Consent Term were collected from the children’s legal representatives. Next, the data collection dynamics began, which took place in two moments (for all years and/or evaluation time). Evaluations of health, physical fitness, real and perceived motor competence, executive function, and physical activity (questionnaire) indicators took place in the first moment. All information was collected during the period of one week (in both shifts—morning and afternoon), on the school premises. In the second moment, the children were asked to use the pedometer to estimate their physical activity. It took 30 days to obtain physical activity data. There was a 12-month interval between assessments for the collection of longitudinal data. The collection of information was carried out by a team of researchers (professors, undergraduate, and Master’s students at the Federal University of Viçosa), trained to carry out the procedures.

### 2.5. Statistical Procedures

Descriptive information was presented as mean and standard deviation, median and interquartile range. The normality of the variables was analyzed using the Shapiro–Wilk test. Student’s *t*-test of independence and Mann–Whitney U test were used to compare gender differences for health, physical activity, physical fitness, real and perceived motor competence, and executive function indicators, according to each cohort. The effect sizes (ES) of the parametric comparisons were computed following the established cutoff points (up to 0.4 small; 0.5 to 0.8 medium; greater than 0.8 large) [46]. Statistical significance was adopted with *p* < 0.05. Analyses were performed using the SPSS version 22 for Windows program (IBM SPSS, Inc., Chicago, IL, USA).

Network analysis was used to assess the association between physical activity, physical fitness components, real and perceived motor competence, and executive function considering gender, WHtR, and BFP for each age within the analyzed cohorts. A network is a collection of nodes and edges, where the nodes represent different variables, and the edges indicate connections between two or more nodes [47]. For example, health indicators, physical fitness components, real and perceived motor competence, and executive functions (inhibitory control, cognitive flexibility, and working memory) represented the nodes in the present study, and the positive and negative relationships between these nodes are the edges. Thus, the role of each variable (node) in the network can be better understood from centrality measures, which are generally used to identify critical areas of the network that can be optimized through intervention processes [47].

Centrality indicators (betweenness, strength, and expected influence) were also reported. Variables with higher betweenness values are more sensitive to change and can act as a hub, connecting other pairs of variables in the network. In other words, betweenness values quantify how often a node is part of the shortest path between all other pairs of nodes connected to the network. The strength indicator is essential for understanding which variables present the most robust connections in the network pattern. Finally, the expected influence is a measure of centrality that takes into account the signal of the weights of the edges—indicating the most influential variables in the network, meaning it provides inference about how influential the nodes are: positive values indicate that the nodes “turn on” the network (i.e., have a positive influence on other nodes), while negative values indicate that the nodes “turn off” the network (i.e., have a negative influence on other nodes). Centrality values were calculated as standardized z-scores to allow comparison between networks.

The components of a system under study are directly influenced depending on how the network is set up, in the same way that the multiple nature of complex systems makes it difficult to identify the “cause” of why factors (different or similar) can lead to different results, depending on the context and history of the individual [26].

The Fruchterman–Reingold algorithm was used. Data were presented in the relative space of the network in which the variables with stronger associations remained together and the less strongly associated variables were repelled from each other [48]. The “random fields of pair-wise Markov” model was used to improve the network accuracy, which was estimated by the “L1” algorithm (regularized neighborhood regression). The regulation was estimated by a less complete selection and contraction operator (Lasso) that has the purpose of controlling the sparse network, and the Extended Bayesian Information Criterion (EBIC) was used [49]. The hyperparameter (y) determines how much EBIC selects sparse models, and therefore the hyperparameter was set to 0.25 (range 0 to 0.50), which is a more parsimonious value in exploratory networks [50]. Furthermore, the network analysis uses regularized absolute minimum contraction and selection operator (LASSO) algorithms in obtaining the precision matrix (weight matrix). The network is a graphical representation that includes variables (nodes) and relationships (edges/lines). Positive relationships are expressed by the blue color and negative relationships are expressed by the red color in the network. The thickness and intensity of the edge indicate the magnitude of the associations [51]. The RStudio software version 4.2.1 program (R Core Team, Vienna, Austria, 2022) and qgraph and ggplot2 packages were used to generate the networks.

## 3. Results

Descriptive information and results of gender comparisons are shown in Table 1. First, 16 girls and 20 boys were evaluated in cohort 1 (Table 1). It was observed that the girls had higher BFG values in the three evaluation moments (T0, t_34_ = 5.07, *p* < 0.001; T1, U = 5.28, *p* < 0.001; T2, U = 67.00, *p* = 0.002). Girls had higher values (steps) for physical activity measured by pedometer than boys in T1, but there was no statistically significant difference. Significant differences were found at T2, in which boys obtained a greater number of steps (t_34_ = −3.52, *p* < 0.001). It was observed that boys had a relatively higher score regarding physical activity obtained through the questionnaire (points) compared to girls for the evaluation moments T0 (t_34_ = −2.17, *p* = 0.037) and T1 (t_34_ = −2.60, *p* = 0.010). Boys showed higher results for physical fitness components compared to girls for abdominal resistance component at T0 (t_34_ = −2.17, *p* = 0.036); speed at T1 (t_34_ = 2.67, *p* = 0.010); and agility at T2 (U = 87.00, *p* = 0.020). Boys also had higher scores than girls regarding motor competence on object control skills in moments T1 (t_34_ = −2.79, *p* = 0.011) and T2 (t_34_ = −2.32, *p* = 0.026).

The sample in cohort 2 consisted of 10 girls and 21 boys (Table 2). Similarly to cohort 1, girls had the highest BFP values in the three evaluation moments (T0, t_29_ = 2.83, *p* = 0.008; T1, U = 23.00, *p* < 0.001; and T2, U = 45.00, *p* = 0.010) when compared to boys. In turn, boys had higher scores on physical activity (questionnaire) at T0 (t_29_ = −2.26, *p* = 0.031) relative to girls. Regarding physical fitness components, girls showed better results for the flexibility component in the three evaluation moments; however, results were statistically significant in T0 (t_29_ = 2.05, *p* = 0.049). Boys showed better results when compared to girls for the components: abdominal resistance at T0 (t_29_ = −2.32, *p* = 0.027) and T1 (t_29_ = −2.17, *p* = 0.038); upper limb strength at T2 (U = 49.00, *p* = 0.017); speed at T1 (t_29_ = 2.48, *p* = 0.019); agility at T0 (t_29_ = 2.97, *p* = 0.006) and T2 (t_29_ = 3.15, *p* = 0.004); and cardiorespiratory fitness at T0 (t_29_ = −2.22, *p* = 0.034). Boys had higher scores for motor competence on object control skills at T1 (U = 49.50, *p* = 0.017) and T2 (t_29_ = −2.31, *p* = 0.028). Furthermore, boys had higher scores at all times for perceived motor competence, being statistically significant at T1 (t_29_ = −2.55, *p* = 0.016). For executive functions, boys had better levels of inhibitory control at T0 (t_29_ =2.19, *p* = 0.036) when compared to girls. In contrast, girls were better than boys in cognitive flexibility at all times, being statistically significant in T1 (t_29_= 2.13, *p* = 0.042).

The topology of the networks of the first cohort is represented in Figure 1. It can be seen that the relationships between variables are arranged differently for each age over time, and a greater number of interactions is observed at eight years of age. No clusters were found between variables at six and seven years old, while health and physical fitness indicators are directly related at eight years old.

Centrality indicators reflect the relative roles of each variable in the network. The centrality indicators of cohort 1 are presented in Figure 2. It is noteworthy that for the strength indicator, the variables: gender, BFP, inhibitory control, and working memory in the six-year-old network; gender, BFP, lower limb strength, and agility in the seven-year-old network; and BFP, speed, agility, and physical activity (number of steps) for the eight-year-old network were higher, being the variables with more robust connections in the pattern of each network. The variables: inhibitory control in the six-year-old network, agility in the seven-year-old network; BFP, upper limb strength, and agility in the eight-year-old network showed the highest betweenness values. It was observed that speed at seven years old, and WHtR and physical activity (number of steps) were the main variables at eight years old regarding the expected influence (meaning they had the highest values).

The networks for cohort 2 are shown in Figure 3. A sparse configuration is observed at eight and 10 years. There was a greater proximity between health, physical fitness, real motor competence, and physical activity indicators at nine years old. Weak relationships can be observed between sociodemographic characteristics and perceived motor competence.

Figure 4 shows the centrality indicators for cohort 2. WHtR and BFP showed the highest values for strength at ages eight, nine and 10, while BFP and cardiorespiratory fitness, and BFP and speed showed the highest values for network connection at eight and nine years old, respectively; there were no betweenness values for the ten-year-old network. It was also observed that the WHtR and lower limb strength variables had the highest values regarding the expected influence indicator in the eight-year-old network; the WHtR, physical activity (points), inhibitory control, and cognitive flexibility variables were the highest in the nine-year-old network; and the WHtR and BFP variables showed higher values in the ten-year-old network.

## 4. Discussion

The study had the main objective to analyze the dynamic and non-linear association through network analysis between health, biological, behavioral, and cognitive variables in children monitored over time, and as a secondary objective, to estimate the gender differences in health, physical activity, physical fitness, real and perceived motor competence, and executive function indicators in three moments.

Results indicated that boys achieved better results for physical fitness, physical activity, and motor competence compared to girls in both evaluation moments (T0 to T2), in both cohorts. Girls had higher values for BFP at all assessment times in both cohorts, and had better results for flexibility and cognitive flexibility ability in the second cohort. The network analysis results indicated different configurations for each age group. Few interactions were observed at six and seven years in cohort 1, and greater interactions between the variables of health, physical activity, real motor competence, and physical fitness at eight years of age. More sparse networks were observed at eight and 10 years of age in cohort 2, and at nine years of age there was a greater interaction between health indicators, motor competence, physical fitness, and physical activity.

Gender differences in physical and motor activities are reported in studies conducted in different countries/cultures [13,20,21,52,53] reporting that boys tend to score higher than girls in a variety of physical and motor tasks [21,54]. Differences between the ages and cohorts analyzed were observed in the present study. Girls had higher BFP values at all ages in both cohorts when compared to boys. These findings in the first cohort may be due to differences in physical activity, nutrition or metabolism, such as the production of the hormone leptin, which may favor an early accumulation of fat in girls, as reported in previous studies [55,56]. Processes related to growth, differences in physical fitness, as well as sexual dimorphism in fat patterning may have occurred in the second cohort [57], as girls showed greater subcutaneous adiposity, which is mainly contributed by the skin fold of the triceps. However, it is highlighted that the findings are worrying due to the possible negative implications for health, and the morbidities (hypertension, diabetes, cardiovascular diseases, development of atherosclerosis, among others) associated with accumulating body fat can compromise the whole development of the child [12,53].

Physical activity behavior (number of steps and questionnaire score) varied between genders according to evaluation times and the studied cohorts. The differences between genders for physical activity were accentuated in younger children, as observed in the literature [13,58], which shows that boys tend to be involved in a greater range of activities than girls of the same age group [52] due to factors such as parental influence and physical activity practice [59]. On the other hand, physical activity behavior did not differ significantly between genders among older children (cohort 2); however, boys had relatively higher values than girls. Such results may be related to self-efficacy (confidence in the ability to be active in specific situations), type of habitual physical activity, unstructured physical activity, and/or active participation in the physical education classes that children were involved in [3]. In addition, the observed results can be attributed to the characteristics of each cohort, in which the learning conditions, the context in which children are involved, and even the age group in which they started the study may have provided this scenario, given that younger children tend to present a more active profile as opposed to older ones.

The results of physical fitness in the present study point to better performance of boys. Boys are encouraged from an early age to participate in more vigorous activities involving greater physical contact, as well as team activities, and outdoor games/play that can contribute to improvements in physical fitness [53,60,61]. In contrast, girls tend to engage in more static, collaborative, and domestic activities, which result in low-intensity activities [61]. Thus, activity patterns and selection throughout childhood are established, which reaffirm higher physical fitness levels among boys [53,62]. However, it is worth mentioning that other factors in recent years such as high exposure to screens (smartphones, tablets, computers, television, etc.), public insecurity, and a lack of leisure spaces can directly impact children’s physical fitness and lifestyle [63].

The motor proficiency of locomotor skills in both genders showed similar values. These results are partly in contrast to the literature, in which there is a tendency for girls to be better in locomotor skills compared to boys [9,52]. However, it is noteworthy that the results are positive for locomotor skills and may contribute to better health trajectories in both genders [30]. On the other hand, boys performed better on object control skills than girls in both cohorts. These findings corroborate previous evidence that observed similar results for object control skills [4,52], in addition to the existence of a relationship between object control skills and physical activity in boys. Evidence supports that girls need other stimuli to guide the practice of physical activity (for example, systematic activities and sports programs) to improve their motor competence, especially manipulative skills [21]. In summary, it is emphasized that boys and girls should be permanently encouraged to get involved in varied activities that stimulate the development and improvement of motor competence [5,64].

Additionally, evidence supports that perceived motor competence changes substantially throughout childhood [29]. Boys had higher perceived motor competence values observed in the second cohort in the present study. This result seems to be in line with previous studies [13,65], which show that the tendency as children get older is to align their perception of motor competence with actual motor competence, which seems to be in line with the real motor competence of the boys (both in locomotor skills and object control), which showed good levels. These findings have positive implications for the developmental trajectories of children to keep them engaged in activities that are challenging, and thus contribute to the adoption of a healthy lifestyle [5,8].

Performance on executive function tasks was different for age and gender. It was observed that younger children (cohort 1) do not differ in terms of gender in their executive functions. Previous research has pointed out that the full development of executive functions seems to occur with greater intensity in early childhood or preschool [66,67], and there is a stabilization of cognitive processes during late childhood, but its assessment is still relevant. In addition, executive functions are positively influenced by physical activity and physical fitness [68], which may be related to improved neural connection, structural and functional brain outcomes, neurogenesis, and release of neurotrophic factors [69]. Boys in the second cohort showed better results for inhibitory control, similar to previous studies [70]. A hypothesis for this difference within the sample may point to the cognitive demand of the task performed, in which boys were quicker to answer the test questions. Inhibition tasks such as reaction time and response accuracy are critical in the developmental process [66], and children with low inhibitory control have difficulty developing responses and paying attention [71], which can negatively impact academic learning and focus on physical and motor activities [15]. Girls showed better results for cognitive flexibility, where potential factors such as differences in motivation, effort, approaches to schoolwork and learning styles, parental expectations, and encouragement may have favored girls [72].

New possibilities for interpreting the associations between health, biological, behavioral, and cognitive variables arise with network analysis. The most robust relationships (which indicate the greatest strength of the network, meaning relationships that were strongly connected with other nodes) in cohort 1 were observed at eight years of age (T2). Based on the topology of the network, a greater commitment to physical activity practice in this phase of life is suggested, aiming at significant contributions to the development of physical fitness [14,73]. It was also observed that physical fitness, specifically the upper limb strength and agility components, together with BFP, acted as a hub in the network at the age of eight due to the greater connectivity with other nodes (variables). Thus, focusing on developing physical fitness [73] may be the best path in future interventions given the connectivity with other health outcomes [20,61].

The WHtR and physical activity variables showed higher expected influence values. From a theoretical perspective, this centrality indicator denotes that these variables are very influential, and better results can generate a positive change in the observed network patterns. From a practical perspective, attention to these variables can shape and plan a given intervention [3,74], meaning that the focus should be on minimizing WHtR values and encouraging children to practice various activities. Given this, the focus on promoting physical activity in this age group is extremely important, as behaviors established during childhood can be perpetuated in adolescence and adult life [74,75]. In addition, WHtR is an indicator of cardiometabolic risk associated with cardiovascular disease, hypertension, and hypercholesterolemia [53]; thus, attention should be paid to this health indicator, especially in childhood.

Cazorla-González et al. [24] recently verified the impact of crawling before walking on network interactions between body composition, cardiovascular system, lung function, motor competence, and physical fitness at seven years of age. The retrospective case-control longitudinal study observed that crawling before walking during child development was a possible modulator in the interaction of networks between body systems at seven years of age, and this skill improved throughout the children’s developmental phase [24]. In general, it was observed that there were greater interactions between physical activity, physical fitness, and health indicators in the present study in cohort 1, but these interactions were different at six (T0), seven (T1), and eight (T2) years of age. However, it is noteworthy that connections with executive functions, real and perceived motor competence, were also observed, but to a lesser extent. Thus, it is assumed that these different interactions may be associated with characteristics resulting from the growth and development process, with direct implications for learning and improving the physical, motor, and cognitive skills of children [16,17,75].

Health indicators showed the greatest strength in all analyzed networks in the second cohort. These findings reinforce the importance of monitoring health indicators (WHtR and BFP) in children. Evidence supports that increased WHtR and BFP indices in childhood tend to perpetuate during adolescence and adulthood [53] and have harmful effects on health and well-being [76]. The greatest interactions were observed between health, physical fitness, physical activity, motor competence, and executive function indicators in the nine-year-old (T1) network. These results may have important practical implications, especially due to the positive association with health and cognitive variables [75]. Encouraging the regular practice of physical activity (whether aerobic, muscle strengthening, or bone strengthening) at this stage of life should be a priority, at least three times a week and 60 min a day [77]. Similar to cohort 1, BFP and cardiorespiratory fitness had high betweenness values. Above all, offering conditions for active practices such as outdoor play, games, structured activities, and active leisure, which enable children to develop and improve their physical fitness levels, can positively impact body adiposity over time [62,73]. Furthermore, WHtR, physical activity, and executive functions (inhibitory control and cognitive flexibility) should not be left out in future interventions as they are very influential variables in the network.

Although the second cohort showed few interactions between investigated variables at eight (T0) years of age, it is noteworthy that the network pattern that emerged was completely different from that observed at eight (T2) years of age in the first cohort. As far as is known, individual characteristics may have contributed to this difference in age overlap; in addition, different activities (or lack thereof) in the after-school shift in both cohorts may have also cooperated with the differences [78]. Even though it is not the focus of the study, the observed results showed that special attention should be given to overlapping ages in future mixed longitudinal studies. Furthermore, few interactions were observed at age 10 (T2). A possible explanation is given to the pre-adolescent period, in which young people’s behaviors change, with reductions in the practice of physical activity associated with a loss of interest in these activities, as well as other adoption of prosocial behavior, making new friendships, etc. [18].

The present study demonstrated different network topologies throughout childhood. These results allow us to infer that the mechanisms behind growth and development are complex, dynamic, and influenced by factors such as time, as children from both cohorts started the research at different ages, and individual (physical and cognitive) and environmental factors should be considered in each age group [26,30]. Thus, integrating several dynamic systems on time scales is not an easy task, and it is even more challenging to be able to investigate this integration during child’s growth and development. This is an important challenge because it opens the door to examining different connections and emerging patterns, as observed in this study. Thus, understanding the complex dynamics through which systems interact over time during childhood makes it possible to create individual and collective interventions (if that is the purpose) which help guide children toward positive health outcomes and developmental trajectories. In summary, most studies used a linear approach when dealing with data, so the comparison of the results observed in this study with other studies was not possible. However, a comprehensive understanding of the investigated phenomena was sought.

The study strengths include direct measures of health, physical activity, physical fitness, motor competence, and executive function indicators; it is a longitudinal study, and it comprises a wide age range—six to 10 years old; it employed the use of complex data analysis—network science; to the best of our knowledge, this is the first study to analyze the dynamic and non-linear association between health, physical activity, physical fitness, real and perceived motor competence, and executive function variables in children over time from a network perspective.

On the other hand, some limitations are highlighted. The larger the sample size, the more stable and accurate the networks are estimated [47]. Thus, the study was limited to a small sample from a specific region of Brazil, and therefore the results should not be generalized. The number of components (variables) in the network system may have changed the accuracy of the networks; however, it is worth highlighting the importance of evaluating different variables associated with human behavior. The differences observed between genders for health, physical activity, physical fitness, real and perceived motor competence, and executive function indicators possibly influenced the interactions observed in networks; however, gender was included (node) in the configuration of networks.

Thus, it seems clear to us that growth and development patterns are different in relation to the specific period that is analyzed and that more longitudinal research focusing on non-linear relationships can be developed in the future. In addition, information about parental behaviors, socioeconomic status, and peer interactions should be considered in future studies.

## 5. Conclusions

The present study showed non-linear dynamic relationships between health, physical activity, physical fitness, real and perceived motor competence, and executive function indicators observed in different network configurations throughout childhood, and identified gender differences for the investigated variables. The relevance of the results indicates that the interactions between the system components can impact children’s development, especially when sociodemographic variables (such as age and gender) are considered. A detailed analysis of network configurations is supported to encourage interventions aimed at promoting physical activity, physical fitness, real and perceived motor competence, executive function, and health indicators of children throughout childhood, with attention to the age at which they presented few interactions.

In addition, the overlapping of ages in mixed-longitudinal studies (although it was not the purpose of the present study) should be reconsidered given the connections established between different variables, directly influencing behavior in childhood. Thus, the present study adds important information to the literature regarding the non-linear, dynamic, and complex relationship between health, biological, behavioral, and cognitive variables of child growth and development over time.

## Figures and Tables

**Figure 1 ijerph-20-02728-f001:**
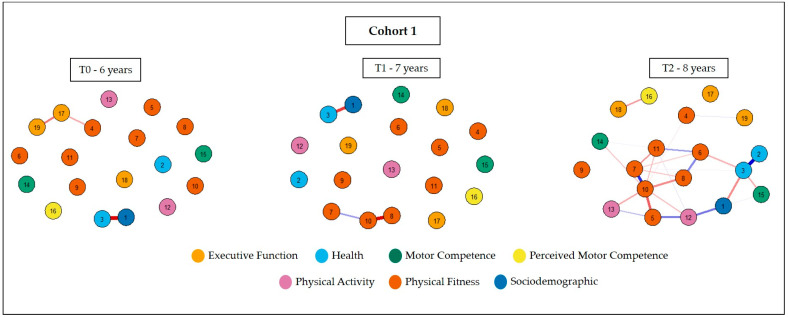
Network of associations between physical activity, physical fitness, real and perceived motor competence, and executive function, considering the sex and health variables of cohort 1. Legend: 1 = sex; 2 = WHtR; 3 = BFP; 4 = hand grip strength; 5 = abdominal resistance; 6 = upper limb strength; 7 = velocity; 8 = lower limb strength; 9 = flexibility; 10 = agility; 11 = cardiorespiratory fitness; 12 = physical activity (steps); 13 = physical activity (score); 14 = locomotion skill; 15 = object control skill; 16 = perceived motor competence; 17 = inhibitory control; 18 = cognitive flexibility; and 19 = working memory.

**Figure 2 ijerph-20-02728-f002:**
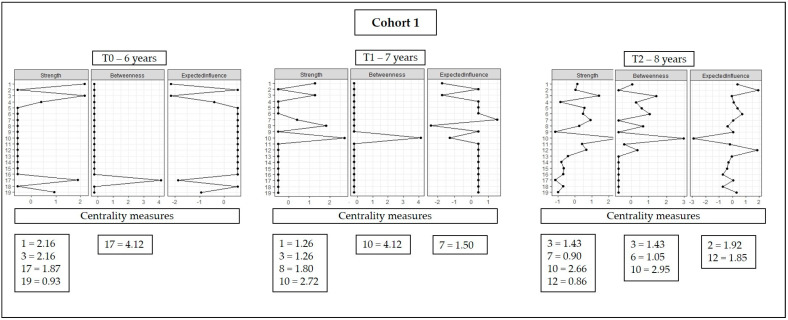
Graphical representation of the centrality indicators of cohort 1.

**Figure 3 ijerph-20-02728-f003:**
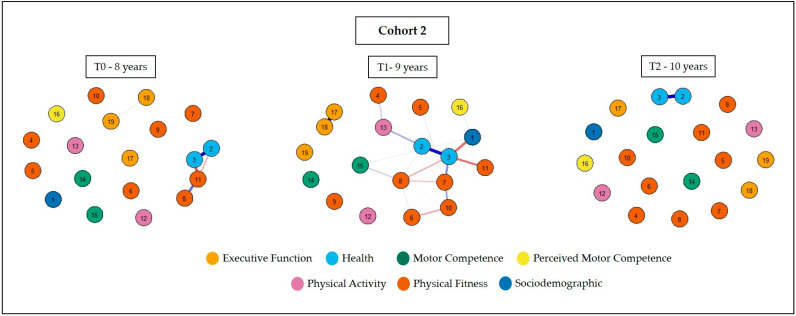
Network of associations between physical activity, physical fitness, real and perceived motor competence, and executive function, considering the sex and health variables of cohort 2. Legend: 1 = sex; 2 = WHtR; 3 = BFP; 4 = hand grip strength; 5 = abdominal resistance; 6 = upper limb strength; 7 = velocity; 8 = lower limb strength; 9 = flexibility; 10 = agility; 11 = cardiorespiratory fitness; 12 = physical activity (steps); 13 = physical activity (score); 14 = locomotion skill; 15 = object control skill; 16 = perceived motor competence; 17 = inhibitory control; 18 = cognitive flexibility; and 19 = working memory.

**Figure 4 ijerph-20-02728-f004:**
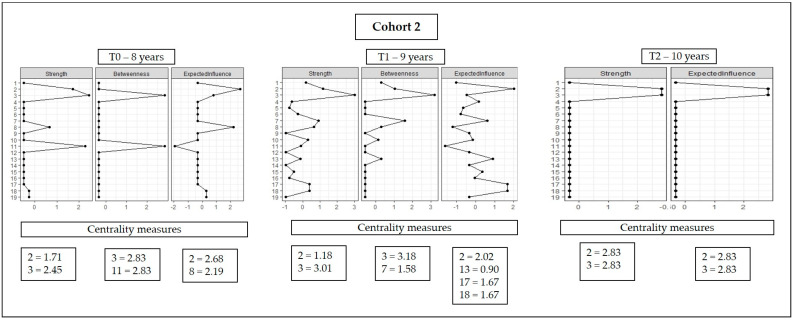
Graphical representation of the centrality indicators of cohort 2.

**Table 1 ijerph-20-02728-t001:** Descriptive information and results for the comparison tests for cohort 1.

	Cohort 1 (Girls = 16; Boys = 20)											
Variables	T0				T1				T2			
	Girls	Boys	t/U	*p*	Girls	Boys	t/U	*p*	Girls	Boys	t/U	*p*
Age (years)	6.31 ± 0.18	6.39 ± 0.97	−0.93 ^a^	0.356 ^c^	7.36± 0.18	7.44 ± 0.27	−1.01 ^a^	0.317 ^c^	8.35 ± 0.18	8.45 ± 0.30	−1.14 ^a^	0.261 ^c^
Health Indicators												
WHtR	0.44 ± 0.02	0.45 ± 0.02	−1.34 ^a^	0.187 ^d^	0.44 ± 0.03	0.44 ± 0.03	−0.28 ^a^	0.782 ^c^	0.44 ± 0.04	0.45 ± 0.03	−0.33 ^a^	0.744 ^c^
BFP (%)	21.31 ± 4.48	13.82 ± 4.34	5.07 ^a^	**<0.001** ^e^	22.16 (6.07)	12.22 (6.87)	5.28 ^b^	**<0.001** ^c^	23.19 (10.31)	12.10 (5.55)	67.00 ^b^	**0.002**
Physical Activity												
Steps	6376.43 ± 2838.16	6916.57 ± 3154.33	−0.53 ^a^	0.597 ^c^	10,322.04 ± 3855.51	8988.12 ± 3044.24	1.16 ^a^	0.254 ^c^	5996.43 ± 2473.21	10,330.91 ± 3154.33	−3.52 ^a^	**<0.001** ^e^
Score	7.38 ± 0.89	8.15 ± 1.16	−2.17 ^a^	**0.037** ^d^	7.67 ± 0.89	8.77 ± 1.48	−2.60 ^a^	**0.010** ^e^	7.83 ± 1.18	8.60 ± 1.35	−1.79 ^a^	0.082 ^d^
Physical Fitness												
Hand grip strength (kg)	9.42 ± 2.10	10.76 ± 2.57	−1.68 ^a^	0.102 ^d^	11.62 ± 2.17	11.40 ± 3.46	0.22 ^a^	0.822 ^c^	12.03 ± 2.32	13.85 ± 4.43	−1.48 ^a^	0.148 ^d^
Abdominal resistance (rep)	8.43 ± 5.07	11.90 ± 4.45	−2.17 ^a^	**0.036** ^d^	12.50 (6.50)	14.00 (3.75)	114.50 ^b^	0.149	13.50 (6.50)	16.00 (6.25)	132.50 ^b^	0.386
Upper limb strength (rep)	2.50 (4.75)	4.10 (7.50)	125.50 ^b^	0.276	3.50 (10.00)	3.00 (7.25)	157.50 ^b^	0.937	3.50 (8.25)	4.50 (8.00)	128.00 ^b^	0.249
Speed (sec)	24.84 (14.01)	25.09 (13.83)	143.00 ^b^	0.604	25.29 ± 1.48	23.78 ± 1.81	2.67 ^a^	**0.010** ^e^	24.22 ± 1.49	23.71 ± 1.67	0.94 ^a^	0.354 ^c^
Lower limb strength (cm)	98.50 (29.43)	95.75 (34.00)	154.00 ^b^	0.863	105.41 (22.65)	119.17 (29.63)	107.50 ^b^	0.095	112.21 ± 15.68	118.29 ± 17.14	−1.09 ^a^	0.280 ^c^
Flexibility (cm)	28.55 ± 5.32	25.26 ± 6.93	1.56 ^a^	0.126 ^d^	27.60 ± 6.78	23.62 ± 7.40	1.66 ^a^	0.103 ^d^	27.28 ± 6.70	23.78 ± 7.71	1.43 ^a^	0.160 ^c^
Agility (sec)	8.68 ± 0.62	8.59 ± 0.72	0.37 ^a^	0.707 ^c^	8.14 (1.18)	7.84 (0.79)	101.00 ^b^	0.062	7.70 (0.75)	7.21 (0.49)	87.00 ^b^	**0.020**
Cardiorespiratory fitness (m)	805.43 ± 141.08	859.56± 135.69	−1.16 ^a^	0.253 ^c^	820.44 ± 114.05	846.27 ± 100.93	−0.72 ^a^	0.483 ^c^	778.01± 143.15	846.31 ± 146.36	−1.40 ^a^	0.169 ^c^
Motor Competence												
Locomotion skill (pts)	35.56 ± 6.12	35.37 ± 4.44	0.10 ^a^	0.916 ^c^	39.12 ± 4.50	38.25 ± 4.29	0.59 ^a^	0.558 ^c^	40.62 ± 4.12	40.00 ± 3.32	0.50 ^a^	0.618 ^c^
Object control skill (pts)	25.18 ± 6.43	27.52 ± 5.56	−1.16 ^a^	0.251 ^c^	29.81 ± 5.10	34.05 ± 4.01	−2.79 ^a^	**0.011** ^e^	36.18 ± 3.35	38.85 ± 3.45	−2.32 ^a^	**0.026** ^e^
Perceived Motor Competence												
Pictographic Scale (pts)	44.50 (8.75)	43.50 (8.75)	131.50 ^b^	0.369	41.50 (10.25)	42.50 (9.75)	126.50 ^b^	0.290	41.00 (10.25)	37.50 (11.75)	128.00 ^b^	0.320
Executive Fuction												
Inhibitory control (sec)	78.00 (32.25)	71.50 (39.50)	120.00 ^b^	0.211	57.50 (26.50)	54.00 (12.50)	114.00 ^b^	0.149	45.00 (18.75)	45.00 (21.25)	149.50 ^b^	0.741
Cognitive flexibility (pts)	3.00 (2.00)	3.00 (1.71)	135.50 ^b^	0.440	4.43 ± 1.71	3.70 ± 1.41	1.41 ^a^	0.166 ^c^	5.00 (1.75)	5.00 (2.75)	156.50 ^b^	0.912
Working memory (pts)	22.00 (4.00)	24.00 (4.25)	159.00 ^b^	0.987	20.00 (4.00)	22.00 (18.00)	155.50 ^b^	0.888	32.50 (11.00)	30.00 (20.00)	143.00 ^b^	0.604

Legend: WHtR—waist-to-height ratio; BFP—body fat percentage; ^a^ t—independence *t* test; ^b^ U—U of Mann–Whitney; ^c^ small effect; ^d^ medium effect; ^e^ large effect; bold value—*p* < 0.05.

**Table 2 ijerph-20-02728-t002:** Descriptive information and results for the comparison tests for cohort 2.

	Cohort 2 (Girls = 10; Boys = 21)											
Variables	T0				T1				T2			
	Girls	Boys	t/U	*p*	Girls	Boys	t/U	*p*	Girls	Boys	t/U	*p*
Age (years)	8.29 ± 0.31	8.19 ± 0.29	0.84 ^a^	0.407 ^c^	9.32 ± 0.33	9.21 ± 0.29	0.85 ^a^	0.400 ^c^	10.31 ± 0.32	10.21 ± 0.29	0.87 ^a^	0.387 ^c^
Health Indicators												
WHtR	0.46 (0.07)	0.43 (0.06)	92.50 ^b^	0.603	0.47 (0.09)	0.43 (0.09)	86.50 ^b^	0.441	0.45 (0.11)	0.43 (0.09)	92.50 ^b^	0.603
BFP (%)	25.12 ± 7.63	16.61 ± 7.90	2.83 ^a^	**0.008** ^e^	29.30 (14.82)	13.10 (12.73)	23.00 ^b^	**<0.001**	22.63 (15.49)	13.69 (14.91)	45.00 ^b^	**0.010**
Physical Activity												
Steps	8204.40 ± 4558.85	9179.80 ± 4908.90	−0.52 ^a^	0.601 ^c^	10,089.92 ± 4516.06	11,580.15 ± 4620.80	−0.84 ^a^	0.405 ^c^	10,264.82 ± 4302.01	11,315.41 ± 5836.02	−0.50 ^a^	0.617 ^c^
Score	7.80 ± 1.33	8.68 ± 0.83	−2.26 ^a^	**0.031** ^e^	8.07 ± 1.03	8.96 ± 1.18	−2.02 ^a^	0.052 ^e^	8.19 ± 1.04	8.49 ± 1.18	−0.69 ^a^	0.491 ^c^
Physical Fitness												
Hand grip strength (kg)	12.00 ± 2.57	12.82 ± 2.67	−0.80 ^a^	0.426 ^c^	14.45 ± 2.94	16.50 ± 3.70	0.65 ^a^	0.137 ^d^	16.95 ± 3.13	17.59 ± 3.65	−0.47 ^a^	0.636 ^c^
Abdominal resistance (rep)	7.10 ± 5.38	12.00 ± 5.54	−2.32 ^a^	**0.027** ^e^	10.40 ± 4.40	14.28 ± 4.75	−2.17 ^a^	**0.038** ^e^	12.10 ± 4.28	14.66 ± 5.42	−1.31 ^a^	0.200 ^d^
Upper limb strength (rep)	4.00 (7.25)	6.00 (7.50)	85.00 ^b^	0.416	2.50 (6.25)	5.00 (7.00)	69.00 ^b^	0.135	1.50 (4.50)	9.00 (8.00)	49.00 ^b^	**0.017**
Spcnhibeecnhibd (sec)	23.94 (1.05)	23.58 (0.95)	65.00 ^b^	0.096	23.97 ± 1.60	22.80 ± 0.99	2.48 ^a^	**0.019** ^e^	23.51 ± 1.33	22.82 ± 1.61	1.16 ^a^	0.253 ^c^
Lower limb strength (cm)	103.49 ± 12.67	115.78 ± 19.05	−1.84 ^a^	0.075 ^d^	112.00 ± 17.76	124.84 ± 21.35	−1.64 ^a^	0.111 ^d^	120.29 ± 11.69	127.11 ± 21.31	−0.94 ^a^	0.354 ^c^
Flexibility (cm)	29.23 ± 3.93	25.71 ± 4.67	2.05 ^a^	**0.049** ^e^	25.81 ± 5.50	24.45 ± 5.52	0.63 ^a^	0.529 ^c^	27.31 ± 5.29	25.99 ± 6.23	1.88 ^a^	0.069 ^c^
Agility (sec)	8.21 ± 0.60	7.56 ± 0.55	2.97 ^a^	**0.006** ^e^	7.36 ± 0.52	7.32 ± 0.42	0.21 ^a^	0.830 ^c^	7.39 ± 0.43	6.86 ± 0.44	3.15 ^a^	**0.004** ^e^
Cardiorespiratory fitness (m)	812.75 ± 68.66	891.91 ± 101.69	−2.22 ^a^	**0.034** ^e^	839.09 ± 120.14	911.00± 95.84	−1.80 ^a^	0.082 ^d^	857.93 ± 67.37	899.02 ± 98.27	−1.19 ^a^	0.243 ^c^
Motor Competence												
Locomotion skill (pts)	36.25 (8.63)	36.50 (7.25)	101.50 ^b^	0.884	39.50 ± 4.74	41.19 ± 2.96	−1.21 ^a^	0.233 ^c^	42.00 ± 2.70	42.61 ± 2.53	−0.62 ^a^	0.539 ^c^
Object control skill (pts)	27.00 ± 5.03	29.40 ± 4.02	−1.43 ^a^	0.162 ^d^	36.00 (6.50)	39.00 (4.00)	49.50 ^b^	**0.017**	38.20 ± 2.57	42.04 ± 4.91	−2.31 ^a^	**0.028** ^e^
Perceived Motor Competence												
Pictographic Scale (pts)	36.10 ± 5.40	39.80 ± 4.94	−1.89 ^a^	0.068 ^d^	36.10 ± 4.45	40.23 ± 4.10	−2.55 ^a^	**0.016** ^e^	36.50 ± 4.35	37.95 ± 4.00	−0.91 ^a^	0.366 ^c^
Executive Fuction												
Inhibitory control (sec)	53.50 ± 9.92	45.76 ± 8.79	2.19 ^a^	**0.036** ^e^	41.00 ± 5.24	37.00 ± 5.33	1.61 ^a^	0.118 ^d^	37.00 (11.00)	35.00 (9.50)	98.00 ^b^	0.787
Cognitive flexibility (pts)	5.00 (2.25)	4.00 (2.00)	72.00 ^b^	0.173	6.60 ± 1.83	5.23 ± 1.57	2.13 ^a^	**0.042** ^e^	6.80 ± 1.47	5.80 ± 2.15	1.30 ^a^	0.202 ^d^
Working memory (pts)	22.00 (6.75)	24.00 (10.00)	102.50 ^b^	0.917	24.00 (8.25)	24.00 (15.00)	95.00 ^b^	0.693	32.50 (12.25)	35.00 (18.00)	87.50 ^b^	0.466

Legend: WHtR—waist-to-height ratio; BFP—body fat percentage; ^a^ t—independence *t* test; ^b^ U—U of Mann–Whitney; ^c^ small effect; ^d^ medium effect; ^e^ large effect; bold value—*p* < 0.05.

## Data Availability

Due to ethics concerns, the data is available upon request to first author.

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
