# Peer review of "How Do Health, Biological, Behavioral, and Cognitive Variables Interact over Time in Children of Both Sexes? A Complex Systems Approach"

_ijerph, 2023, doi:10.3390/ijerph20032728_

Round 1

Reviewer 1 Report

ReviewIJERPH-2125949

Article title - How do health, biological, behavioral, and cognitive variables interact over time in children? A complex systems approach

Overview

            With a longitudinal method, the paper addresses a well-known issue in the field of sports and health sciences and is helpful for addressing intra- and inter-individual developmental questions. Additionally, it suggests using networks to characterize and consider these potential changes in children between the ages of 6 and 10.

            The current text has flaws, as a scientific article is expected to be direct, objective, and theoretically consistent. Here are some observations and recommendations for authors.

Title

·      The title must be changed to reflect your content. The study and discussion of gender differences are fundamental to the manuscript. This information must be included in the title.

·      The title also implies that there would be a theoretical insertion using concepts and principles of complex systems. However, this does not occur, demonstrating yet another lack of coherence between the title and the content of the article. It is recommended to avoid this theoretical "flag" in the title.

·      When revising the title of a scientific study, authors should think about how to be as clear, precise, and succinct as possible.

Introduction

To clearly demonstrate how the study questions and objectives arise, it appears that the Introduction needs to be adjusted.

·      The crucial point is not adequately explained or supported by any references in the sentence between lines 60 and 64.

“A general point in common among the works that investigated the relationships between physical activity, real and perceived motor competence, physical fitness, executive function, and health indicators in children is to point out the differences between the genders, in addition to considering that these relationships are established linearly, disregarding dynamic and non-linear synergistic interactions between variables”

This sentence needs to be rewritten by the writers to make it more obvious what they were trying to say.

·      The cross-sectional studies [20,21] referenced to support the study's justification looked at subjects at a different developmental stage (preschoolers), with variables distinct from those utilized in the present study. These differences should be properly noted to prevent confounding the reader. However, the study by Cazorla-González et al. [22] should be more thoroughly discussed in the Introduction because it is more relevant to the current investigation, employs a longitudinal network analysis, uses 7-year-old children, and has many of the same variables (body composition, the cardiovascular system, motor competence, physical fitness, and physical activity).

·      It is important to clarify the research problem and the primary and secondary goals in the Introduction. The Introduction indicated that there are two objectives that are equally important, and the authors should reconsider this. The study appears to give the Results and Discussion associated with the first objective, which is related to the investigation of gender differences. The second objective, which concerns the longitudinal study of the data using network analysis, is not explicitly linked to the first one, which ultimately leads to an article that offers two distinct investigations. The authors may give a network analysis that takes gender differences into account to break this impasse.

Methods.

·      The study is extremely vulnerable due to several information gaps that make it impossible to determine the criteria for generalizing the data, the veracity of the data that was obtained, or the proper explanation of specific techniques. Authors are required to address the following flaws: lack of details regarding the type and format of the sampling employed in the study; lack of information from studies that validated the data collection methods; lack of information on sample size and power estimation.

·      The fact that the network of correlations between the variables included in this study does not arise from the time points under consideration seems more serious. This raises the question, do these networks actually exist? Or was the development of the network compromised by some significant methodological flaw, such as the sample size, which was not estimated for the number of associations to be evaluated (see Cazorla-González et al. [22])? To make the manuscript more comprehensible and objective, the authors must take this into account and make the appropriate revisions.

Discussion

It is likely that the Discussion will need to be adjusted in proportion to the changes asked for in the earlier parts.

Additional queries

·      The variables referred to as "Health" are those related to weight status. This is because the variables "Health" also include "Physical Activity" and "Physical Fitness". This should be changed in the complete document, please.

·      It would be better to keep with the description of the time of collection T0, T1, and T2 in Figures 1, 2, 3, and 4, as the reader might become confused by the ages of the subjects.

·      The sample's "n" in Tables 1 and 2 must be provided.

·      Reference [63] in line 460 has already been supplanted by more current references. Please, update.

·      Line 535 - What do values of "intermediation" mean? Is there a problem with the translation here?

Final view

One of the key comments on this study is the fact that this article offers a thorough and in-depth analysis of the gender differences in various active and healthy lifestyle variables. The perspective of the networks of associations between these variables is also very appealing. However, significant limitations on sample size (and non-probabilistic sampling) might make it impossible for the authors to deliver the network analysis over time.

The gap shown by this paper is noteworthy since it combines the work of other researchers to investigate additional and better ways to represent changes in human development. This article must work for its theoretical and methodological solidity because other researchers will be able to be inspired by it.

Reviewer 2 Report

Dear authors,

I have analyzed your research with great interest. The manuscript is very well structured and has a high level of complexity, due to the multiple variables analyzed. You have stated the main limitation of the study (related to the investigated sample), so I can only suggest a few ideas for improving the current version of your work:

1. The title refers only to the second specified research direction (lines 88-90). The first line of research (relating to gender differences for the investigated parameters/lines 86-88) is not indicated in the title of the article.

2. Lines 97-99: This longitudinal study is part of the research project “Relationship between Physical 97 Activity, Motor Competence, Cognitive Skills and School Performance in Children and 98 Adolescents from three to 12 years old. The two investigated samples/groups include only children aged between 6-8 and 8-10 years, respectively, according to the information presented (lines 115-117). Do you intend to use the data obtained from the 3-6 years and 10-12 years groups in other publications?

3. Tables 1 and 2: For parametric data where you used the t-test (parametric) for independent samples, it may be useful to add the effect size (Cohen's d) values. It is only a suggestion, the research results are clearly enough presented in the current version.

4. It would have been interesting to analyze the evolution of the results for the 3 assessments, by comparing the differences separately, for each gender (Anova with repeated measurements for the parametric variant, respectively the Friedman test for the non-parametric variant). The volume of data in the current version is very large, so you could present it in other publications.

5. Table 1 (Physical Fitness / Cardiorespiratory fitness / Girls): T0= 805.43±141.08 m, T1= 820.44±114.05m, T2= 778.01± 143.15. How do you explain this apparent decrease between T1 and T2 values?

6. The results in tables 1 and 2 present the mean/average and SD for the variables where the normality of the data distribution is ascertained and the t-test was used. For the non-parametric version (where the Mann-Whitney U test was used) is the Mean Rank presented? This aspect can be confusing for those who follow/read the evolution of results over time/horizontally (T0-T1-T2). For example, in table 2/ Velocity (seconds): T0=12.35(4.00)/girls and 12.00(1.13)/boys for non-parametric, T1= 23.97±1.60/girls and 22.80±0.99/boys for parametric, T2= 23.51± 1.33/girls and 22.82±1.61/boys for parametric.
